# The Emergence of Sars-CoV-2 Variant Lambda (C.37) in South America

Pedro E. Romero,[a,b] Alejandra Dávila-Barclay,[a] Guillermo Salvatierra,[a] Luis González,[a] Diego Cuicapuza,[a] Luis Solís,[a] Pool Marcos-Carbajal,[a,f] Janet Huancachoque,[a] Lenin Maturrano,[c] Pablo Tsukayama[a,d,e]

aLaboratorio de Genómica Microbiana, Facultad de Ciencias y Filosofía, Universidad Peruana Cayetano Heredia, Lima, Peru

bLaboratorio de Biología Marina, Facultad de Ciencias y Filosofía, Universidad Peruana Cayetano Heredia, Lima, Peru

cLaboratorio de Microbiología, Facultad de Medicina Veterinaria, Universidad Nacional Mayor de San Marcos, Lima, Peru

dInstituto de Medicina Tropical Alexander von Humboldt, Lima, Peru

eWellcome Sanger Institute, Hinxton, United Kingdom

fEscuela Profesional de Medicina Humana, Universidad Peruana Unión, Lima, Peru

**KEYWORDS** C.37, COVID-19, SARS-CoV-2, South America, genome analysis

The evolution of SARS-CoV-2 variants with potentially increased transmissibility, virulence, and resistance to antibody neutralization poses new challenges for the control of COVID-19 (1), particularly in low- and middle-income countries (LMICs) where transmission remains high and vaccination progress is still incipient.

Peru has been severely hit by the COVID-19 pandemic: as of 6 October 2021, it had the highest rate of COVID-19 deaths globally relative to its population (199,520 out of 33.57 million: 0.59% of the country's population) (https://www.gob.pe/institucion/minsa/informes-publicaciones/1944190-criterios-tecnicos-para-actualizar-la-cifra-de-fallecidos-por-covid-19-en-el-peru). By July 2021, 3,100 genome sequences from Peru were available on GISAID, comprising 60 circulating PANGO lineages (https://nextstrain.org/community/quipupe/Peru_lambda). Routine genomic surveillance in early 2021 revealed a deep-branching sublineage of B.1.1.1, now classified as C.37 (Fig. 1A). It was first reported in Lima in December 2020 (1 of 216 genomes, 0.5%), expanding to 21.7%, 29.9%, 46.4%, 90.2%, 72.6%, and 82.2% in January, February, March, April, May, and June 2021, respectively (Fig. 1B). In contrast, variants of concern were detected less frequently over these 6 months in Peru: alpha, $n = 23$, 0.7%; gamma, $n = 350$, 11.3%.

C.37 contains a novel deletion (S: Δ246 to 252, located at the N-terminal domain) plus seven nonsynonymous mutations in the Spike gene (G75V, T76I, D253N, L452Q, F490S, D614G, T859N) (Fig. 1C). Mutations L452Q and F490S both map to the Spike protein's receptor-binding domain (RBD). While L452Q is almost exclusive to C.37, L452R is present in variant of concern (VOC) delta (B.1.617.2) and variants of interest (VOI) epsilon (B.1.427/B.1.429) and kappa (B.1.617.1) and is associated with increased affinity for the ACE2 receptor (2). F490S has been associated with reduced *in vitro* susceptibility to antibody neutralization (3, 4). C.37 also displays the ORF1a Δ3675 to 3677 deletion, found in VOCs alpha, beta, and gamma (5).

The earliest record of C.37 on GISAID is from Argentina in November 2020. By 6 October 2021, there were 7,706 C.37 sequences from 55 countries, including Peru ($n = 3,126$), Chile ($n = 1,780$), USA ($n = 181$), Mexico ($n = 213$), Argentina ($n = 467$), Ecuador ($n = 254$), Spain ($n = 223$), Germany ($n = 101$), Colombia ($n = 79$), and France ($n = 61$). Beyond Peru, C.37 expanded rapidly in Chile and Argentina, reaching up to 40% and 36% of all sequenced genomes on GISAID by July 2021, respectively (Fig. 1B). The emergence of this lineage in Peru and its export to other countries is a current hypothesis, given its earlier detection and rise to nearly 90% of sequences by April. We are processing additional Peruvian samples from October to December 2020 to confirm and date the origin of C.37.

Address correspondence to Pablo Tsukayama, pablo.tsukayama@upch.pe.

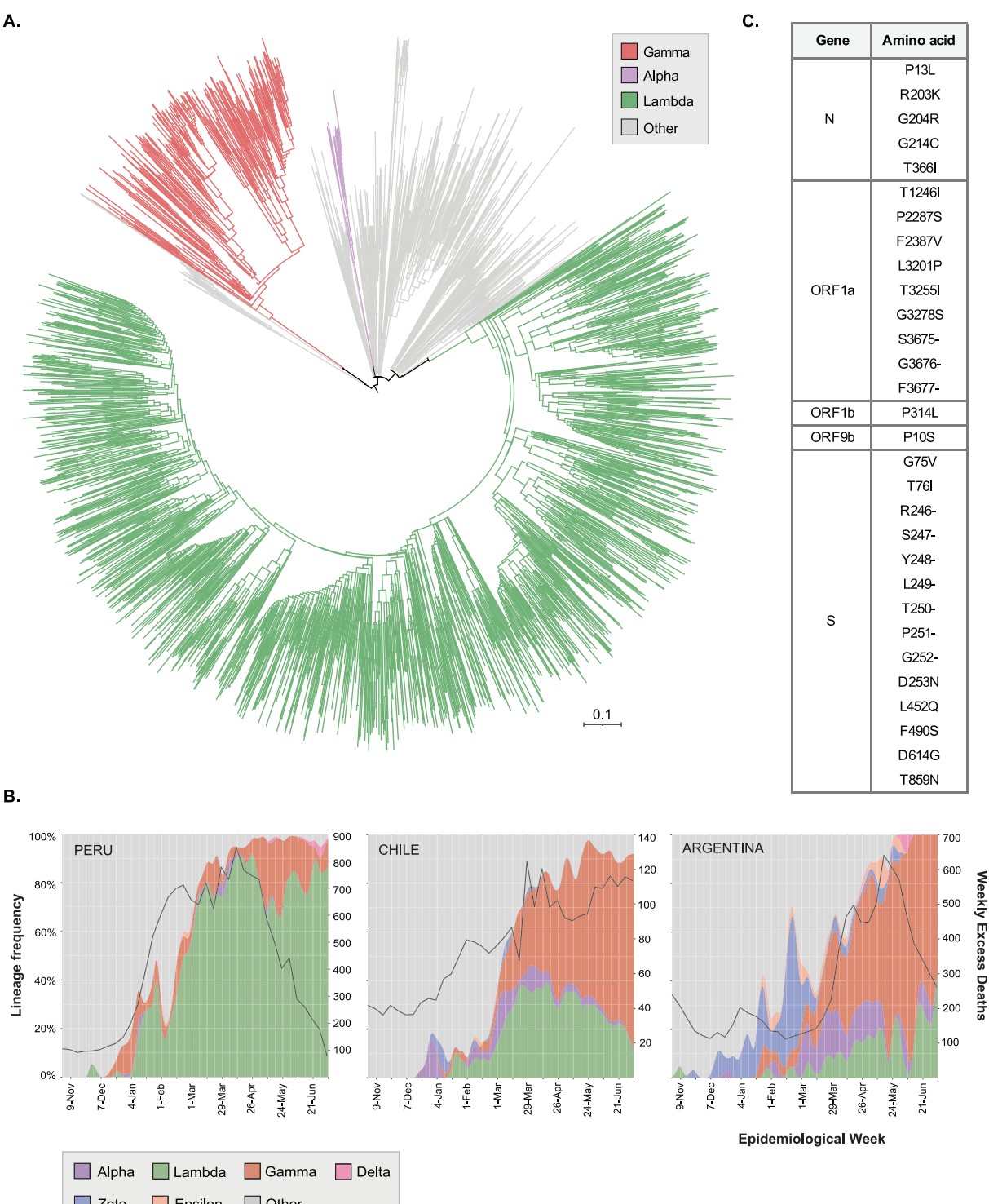

**FIG 1** (A) Maximum likelihood tree from 3,100 SARS-CoV-2 genomes reported in Peru between 1 November 2020 and 1 July 2021, highlighting variants alpha (*n* = 23), gamma (*n* = 350), and lambda (*n* = 2,077). The tree was obtained using the *augur* pipeline (6) under a general time-reversible (GTR) model of nucleotide substitution and assuming a clock rate of $8 \times 10^{-4}$. The scale represents the number of mutations. Complete information of other circulating PANGO lineages can be accessed from https://nextstrain.org/community/quipupe/Peru_lambda. Two genomes from China (EPI_ISL_402123, EPI_ISL_406798) were used as the outgroup. (B) Relative frequencies of predominant SARS-CoV-2 variants in public genomic data sets from Peru (*n* = 3,100), Chile (*n* = 4,469), and Argentina (*n* = 1,331) from November 2020 to July 2021. The black curve indicates excess mortality deaths relative to previous years by epidemiological week, obtained from the coronavirus R package (https://github.com/RamiKrispin/coronavirus). (C) Nonsynonymous mutations present in SARS-CoV-2 variant lambda.

Expansion of C.37 has occurred in South America in the presence of hundreds of circulating lineages and VOCs alpha and gamma (Fig. 1B), suggesting increased transmissibility of this lineage relative to that of the parental Wuhan strain. However, additional epidemiological data and analyses are needed to assess its transmission, virulence, and immune escape properties.

On 15 June 2021, the World Health Organization designated C.37 as VOI lambda (https://www.who.int/publications/m/item/weekly-epidemiological-update-on-covid-19—15-june-2021).

All analyzed sequences were publicly available in GISAID at the time of manuscript submission. Raw Illumina reads from all Peruvian genomes sequenced at UPCH are available at NCBI BioProject PRJNA667090. A list of authors and related metadata from all sequences included in our analyses can be downloaded from https://nextstrain.org/community/quipupe/Peru_lambda. All code and data have been posted to a dedicated Git repository (https://github.com/LGM-UPCH/peru_lambda).

## ACKNOWLEDGMENTS

The Institutional Review Board of Universidad Peruana Cayetano Heredia approved the project in June 2020 (E051-12-20). We are funded by Fondo Nacional de Ciencia y Tecnología (FONDECYT) grants 046-2020 and 022-2021 and Universidad Nacional Mayor de San Marcos grant A2008007M. P.E.R. is supported by FONDECYT grant 034-2019. We thank Instituto Nacional de Salud and many collaborators in Peru for providing clinical specimens for sequencing. Computational experiments were performed at Centro de Alto Rendimiento Computacional del Instituto de Investigaciones de la Amazonía Peruana (IIAP). We acknowledge colleagues in the laboratories that generated and shared genetic sequence data via the GISAID Initiative, on which this study is based.

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
