## [Reviewer comments · Microbiology Spectrum]

Microbiology Spectrum

The Emergence of SARS-CoV-2 Variant Lambda (C.37) in South America

Pedro Romero, Alejandra Dávila-Barclay, Guillermo Salvatierra, Luis González, Diego Cuicapuza, Luis Solís, Pool Marcos-Carbajal, Janet Huancachoque, Lenin Maturrano, and Pablo Tsukayama

Corresponding Author(s): Pablo Tsukayama, Universidad Peruana Cayetano Heredia

Review Timeline:

Submission Date:	July 11, 2021
Editorial Decision:	September 7, 2021
Revision Received:	October 9, 2021
Accepted:	October 11, 2021

Editor: Heba Mostafa

Reviewer(s): Disclosure of reviewer identity is with reference to reviewer comments included in decision letter(s). The following individuals involved in review of your submission have agreed to reveal their identity: Peter Thielen (Reviewer #1); Amary Fall (Reviewer #3)

Transaction Report:

DOI: <https://doi.org/10.1128/Spectrum.00789-21>

September 7, 2021

Dr. Pablo Tsukayama
Universidad Peruana Cayetano Heredia
Department of Microbiology
Av Honorio Delgado 430
Lima 15102
Peru

Re: Spectrum00789-21 (The Emergence of SARS-CoV-2 Variant Lambda (C.37) in South America)

Dear Dr. Pablo Tsukayama:

Thank you for submitting your manuscript to Microbiology Spectrum. As you will see your paper is very close to acceptance. Please modify the manuscript along the lines I have recommended. As these revisions are quite minor, I expect that you should be able to turn in the revised paper in less than 30 days, if not sooner. If your manuscript was reviewed, you will find the reviewers' comments below.

When submitting the revised version of your paper, please provide (1) point-by-point responses to the issues I raised in your cover letter, and (2) a PDF file that indicates the changes from the original submission (by highlighting or underlining the changes) as file type "Marked Up Manuscript - For Review Only". Please use this link to submit your revised manuscript. Detailed information on submitting your revised paper are below.

Link Not Available

Sincerely,

Heba Mostafa

Reviewer comments:

Reviewer #1 (Comments for the Author):

This manuscript provides a very concise description of the C.37 variant, now referred to as Lambda, as it emerged and became prevalent in Peru and South America. The authors describe the defining mutations of the virus as well as its observed expansion within the geographic region using data generated by the international community and their own work. Overall, this is a nice description of the emergence of the new variant, and it would serve as a good reference for future studies aiming to evaluate the

Comments:

- 1) This paper seems to be missing a methods section. There is new data generated and analyzed, but no description of the processes used to generate it, reference of institutional approvals for the study, etc. Consider expanding this section based on the journal requirements - I have not looked deeply in to this submission type to determine how detailed this section is intended to be, but I appreciate being able to quickly understand how data were produced and what git repos were used to analyze data.
- 2) Is it realistic to maintain a NextStrain website indefinitely? Currently, reference 3 is very useful for exploration. However, in the future (say, a year or more from now) I anticipate this will be confusing to a reader given increased data or the site no longer being active. Consider adding an accessed date, or making this specific reference static.

3) Data submissions. Consider submitting data to genbank. Also, the note for the SRA indicates 350 samples exist, but on looking at the list that isn't true. Make sure these numbers match.

4) GISAID submissions. Unclear if this has recently changed, but acknowledging GISAID submitters usually requires listing them in a supplemental table. Overall, it would be useful to have a list of accessions you used for your reference dataset so it would be easier to reproduce by others.

Reviewer #3 (Comments for the Author):

This is an interesting study on the Emergence of SARS-COV 2 variant lambda (c.37) in South America. The manuscript is an interesting read, but twice points that need to be reviewed by the authors.

1) There is an inconsistency between the abstract and the text with respect to the number nonsynonymous mutation number.

2) What was the globally rate of COVID-19 deaths at this moment?

Preparing Revision Guidelines

- point-by-point responses to the issues I raised in your cover letter
- Upload a compare copy of the manuscript (without figures) as a "Marked-Up Manuscript" file.
- Each figure must be uploaded as a separate file, and any multipanel figures must be assembled into one file.
- Manuscript: A .DOC version of the revised manuscript
- Figures: Editable, high-resolution, individual figure files are required at revision, TIFF or EPS files are preferred

Please return the manuscript within 60 days; if you cannot complete the modification within this time period, please contact me. If you do not wish to modify the manuscript and prefer to submit it to another journal, please notify me of your decision immediately so that the manuscript may be formally withdrawn from consideration by Microbiology Spectrum.

This manuscript provides a very concise description of the C.37 variant, now referred to as Lambda, as it emerged and became prevalent in Peru and South America. The authors describe the defining mutations of the virus as well as its observed expansion within the geographic region using data generated by the international community and their own work. Overall, this is a nice description of the emergence of the new variant, and it would serve as a good reference for future studies aiming to evaluate the

Comments:

- 1) This paper seems to be missing a methods section. There is new data generated and analyzed, but no description of the processes used to generate it, reference of institutional approvals for the study, etc. Consider expanding this section based on the journal requirements – I have not looked deeply in to this submission type to determine how detailed this section is intended to be, but I appreciate being able to quickly understand how data were produced and what git repos were used to analyze data.
- 2) Is it realistic to maintain a NextStrain website indefinitely? Currently, reference 3 is very useful for exploration. However, in the future (say, a year or more from now) I anticipate this will be confusing to a reader given increased data or the site no longer being active. Consider adding an accessed date, or making this specific reference static.
- 3) Data submissions. Consider submitting data to genbank. Also, the note for the SRA indicates 350 samples exist, but on looking at the list that isn't true. Make sure these numbers match.
- 4) GISAID submissions. Unclear if this has recently changed, but acknowledging GISAID submitters usually requires listing them in a supplemental table. Overall, it would be useful to have a list of accessions you used for your reference dataset so it would be easier to reproduce by others.

Manuscript Spectrum00789-21R1: Response to Reviewers

Reviewer #1

This manuscript provides a very concise description of the C.37 variant, now referred to as Lambda, as it emerged and became prevalent in Peru and South America. The authors describe the defining mutations of the virus as well as its observed expansion within the geographic region using data generated by the international community and their own work. Overall, this is a nice description of the emergence of the new variant, and it would serve as a good reference for future studies aiming to evaluate the

1) This paper seems to be missing a methods section. There is new data generated and analyzed, but no description of the processes used to generate it, reference of institutional approvals for the study, etc. Consider expanding this section based on the journal requirements - I have not looked deeply in to this submission type to determine how detailed this section is intended to be, but I appreciate being able to quickly understand how data were produced and what git repos were used to analyze data.

Thank you. This manuscript was submitted as a New Data Letter article with a 500-word limit. We summarize the methods in the figure legend and set up a Git repository that expands on the methods and lists the sequences and code used to reproduce the analysis. It can be found at: https://github.com/LGM-UPCH/peru_lambda, and this has been added to the Data Availability section at the end of the manuscript.

2) Is it realistic to maintain a NextStrain website indefinitely? Currently, reference 3 is very useful for exploration. However, in the future (say, a year or more from now) I anticipate this will be confusing to a reader given increased data or the site no longer being active. Consider adding an accessed date, or making this specific reference static.

We have created a new Nexstrain build that includes the 3100 Peruvian genomes analyzed in the manuscript (between November 2020 and July 2021). This build will not be updated further and will remain public indefinitely. The URL has been updated in Ref. 3.

3) Data submissions. Consider submitting data to genbank. Also, the note for the SRA indicates 350 samples exist, but on looking at the list that isn't true. Make sure these numbers match.

We have uploaded 544 fastq sets to SRA-Genbank, corresponding to all isolates sequenced in our laboratory since March 2020. The assemblies are already in GISAID. Numbers have been updated in the manuscript text.

4) GISAID submissions. Unclear if this has recently changed, but acknowledging GISAID submitters usually requires listing them in a supplemental table. Overall, it would be useful to

have a list of accessions you used for your reference dataset so it would be easier to reproduce by others.

We could not include the list of authors as a supplemental table due to space constraints. However, the complete list of GISAID accessions, plus author information and all metadata provided by the originating labs, can be downloaded from the Nexstrain build in Ref. 3. This has been included in the Data availability section in the manuscript.

Reviewer #3

This is an interesting study on the Emergence of SARS-COV 2 variant lambda (c.37) in South America. The manuscript is an interesting read, but twice points that need to be reviewed by the authors.

1) There is an inconsistency between the abstract and the text with respect to the number nonsynonymous mutation number.

Thank you. This has been updated in the text and is now set as eight nonsynonymous mutations (including the 246-252 deletion) in the S gene.

2) What was the globally rate of COVID-19 deaths at this moment?

We have addressed this point in Figure 1B by overlaying (publicly available) curves on excess deaths relative to previous years for all three countries. This is our most direct way to compare deaths rates across the region over the analysis period, and suggests an association between the arrival of Lambda and Variants of Concern in early 2021.

October 11, 2021

Dr. Pablo Tsukayama
Universidad Peruana Cayetano Heredia
Department of Microbiology
Av Honorio Delgado 430
Lima 15102
Peru

Re: Spectrum00789-21R1 (The Emergence of SARS-CoV-2 Variant Lambda (C.37) in South America)

Dear Dr. Pablo Tsukayama:

Your manuscript has been accepted, and I am forwarding it to the ASM Journals Department for publication. You will be notified when your proofs are ready to be viewed.

Sincerely,

Heba Mostafa
Editor, Microbiology Spectrum
